# Epipolar-Free 3D Gaussian Splatting for Generalizable Novel View Synthesis

**Zhiyuan Min**[1] **Yawei Luo**[1,*] **Jianwen Sun**[2] **Yi Yang**[1]

[1]Zhejiang University [2]Central China Normal University

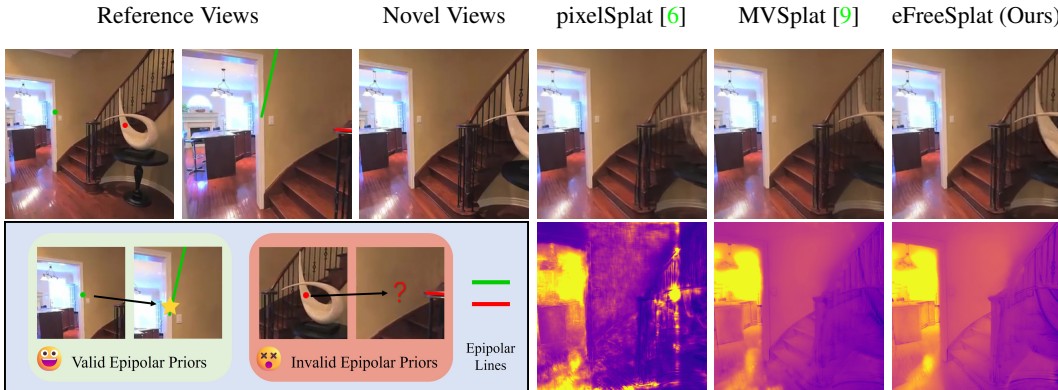

**Fig. 1.** Epipolar priors can be unreliable across extremely sparse views, especially in non-overlapping or occluded areas. Our model, eFreeSplat, generalizes to novel scenes without relying on epipolar priors, offering superior appearance and geometric perception.

## Abstract

Generalizable 3D Gaussian splitting (3DGS) can reconstruct new scenes from sparse-view observations in a feed-forward inference manner, eliminating the need for scene-specific retraining required in conventional 3DGS. However, existing methods rely heavily on epipolar priors, which can be unreliable in complex real-world scenes, particularly in non-overlapping and occluded regions. In this paper, we propose eFreeSplat, an efficient feed-forward 3DGS-based model for generalizable novel view synthesis that operates independently of epipolar line constraints. To enhance multiview feature extraction with 3D perception, we employ a self-supervised Vision Transformer (ViT) with cross-view completion pre-training on large-scale datasets. Additionally, we introduce an Iterative Cross-view Gaussians Alignment method to ensure consistent depth scales across different views. Our eFreeSplat represents an innovative approach for generalizable novel view synthesis. Different from the existing pure geometry-free methods, eFreeSplat focuses more on achieving epipolar-free feature matching and encoding by providing 3D priors through cross-view pretraining. We evaluate eFreeSplat on wide-baseline novel view synthesis tasks using the RealEstate10K and ACID datasets. Extensive experiments demonstrate that eFreeSplat surpasses state-of-the-art baselines that rely on epipolar priors, achieving superior geometry reconstruction and novel view synthesis quality. Project page: https://tatakai1.github.io/efreesplat/.

---

*Corresponding author

38th Conference on Neural Information Processing Systems (NeurIPS 2024).

# 1 Introduction

Rendering novel views from sparse observations has long been a challenging research task in the 3D vision community. Recently, generalizable novel view synthesis (GNVS) techniques have emerged as a promising solution. These models, trained on large-scale multiview datasets, can directly synthesize novel views of new scenes from a few observations, eliminating the need for scene-specific retraining. Notable works in this vein include NeRF-based GNVS [37, 38, 55, 64] and Light Field Network-based GNVS [12, 48, 49]. An enabling factor in their generalizability is the use of epipolar priors, which help determine the precise location of a pixel in one image on the corresponding epipolar line in another viewpoint [17, 70]. More recently, generalizable 3D Gaussian splatting methods, such as pixelSplat [6] and MVSplat [9], have been proposed. These methods leverage the benefits of a primitive-based 3D representation, offering fast and memory-efficient rendering along with an interpretable 3D structure for generalizable view synthesis. Like previous approaches, most 3DGS-based GNVS methods [6, 9, 61] depend on epipolar priors to achieve high-quality and fast cross-scene novel view rendering.

Despite significant advancements utilizing epipolar priors, a new and underexplored issue has emerged in GNVS: epipolar priors prove unreliable in non-overlapping and occluded regions of complex real-world scenes, where corresponding points on epipolar lines are absent. As depicted in Fig. 1, epipolar lines (marked in green) effectively identify geometric correspondences in multiview overlapping areas. Conversely, epipolar lines (marked in red) become invalid in those non-overlapping regions, leading to unreliable geometric reconstructions. Moreover, sampling on invalid epipolar lines and employing attention mechanism will produce a lot of redundant calculations [6, 38, 49].

A newly proposed geometry-free 3D reconstruction method [56], which captures multiview consistent knowledge from a versatile model pre-trained on cross-view data, has inspired our development of a novel GNVS method that circumvents the dependence on epipolar priors through data-driven 3D priors. Leveraging this insight, we propose eFreeSplat, an efficient feed-forward 3D Gaussian Splatting model for GNVS that operates independently of epipolar line priors. eFreeSplat is built upon 3DGS [23] originally designed for single-scene NVS and extends its advantages to GNVS. The overview of our method is illustrated in Fig. 2. To capture 3D structural information across sparse views without unreliable epipolar priors, we utilize a self-supervised pre-training model for 3D cross-view completion [59, 60]. This model uses a Vision Transformer (ViT) [11] encoder and cross-attention decoder to predict parts of the masked images from reference views. In eFreeSplat, the pre-training model retains all patches, effectively capturing spatial relationships and acting as a "*cross-view mutual perceiver*". This approach provides robust geometric biases for global 3D representation via cross-view completion pre-training on large-scale datasets [25, 35, 40, 44, 45].

Experimentally, we found that without an explicit 3D constraint, the scale of predicted depth maps of per-pixel 3D points from different views tends to be inconsistent [4, 53], leading to artifacts or pixel displacement in images from novel views. To address the issue of inconsistent depth scales across different views, we introduce an Iterative Cross-view Gaussians Alignment (ICGA) technique to eFreeSplat. ICGA is based on the fact that the features of most surface points projected onto the camera planes of different views remain consistent. Specifically, we obtain the warped features for each view based on the predicted depths via U-Net. We then calculate the fine depths for the next iteration via the correlation between the warped features and the features from other views. Unlike the plane-sweep stereo approach [9, 62, 63], our updating and alignment strategy does not require numerous depth candidates, thereby reducing computational and storage costs.

The main contributions of this paper are summarized as follows:

- We introduce eFreeSplat, a method with novel insights into GNVS that operates without relying on epipolar priors in the process of multi-view geometric perception. eFreeSplat demonstrates robustness in generalizing to new scenarios with sparse and non-overlapping observations.

- To ensure depth scale consistency across different viewpoints without explicit epipolar constraints, we propose an Iterative Cross-view Gaussians Alignment method, which alleviates artifacts and pixel displacement issues in renderings.

- eFreeSplat achieves competitive cross-scene rendering performance on the RealEstate10K [72] and ACID [26] datasets, surpassing state-of-the-art approaches such as pixelSplat [6] and MVSplat [9].

## 2 Related Work

**Single-Scene 3DGS.** 3D Gaussian Splatting (3DGS) [23] marks a significant shift in 3D scene representation. It employs millions of learnable 3D Gaussians to explicitly map spatial coordinates to pixel values, enhancing rendering efficiency and quality via a rasterization-based splatting approach, and boosting various downstream tasks [34, 36]. Unlike early 3D neural representation methods [37, 39, 46] that require intensive computations and large memory usage (*e.g.*, neural fields [2, 3, 67] and volume rendering [27, 65, 66]) , 3DGS enables real-time rendering and editability with minimized computational demands [8]. Existing single-scene 3DGS-liked methods [10, 18, 23] demand dense views for each scene via the expensive per-scene gradient back-propagation process. In our work, we employ a single feedforward network to deduce the parameters of Gaussian primitives using merely two images.

**Cross-Scene Generalizable 3DGS.** Cross-scene generalizable 3DGS learns robust priors from large-scale scenarios to predict Gaussian primitive parameters and render novel view images using sparse inputs. pixelSplat [6] and LatentSplat [61] leverage the epipolar transformer [17] to find cross-view correspondences and learn per-pixel Gaussian depth distributions. However, this can fail in non-overlapping and occluded areas, leading to inaccurate geometry and surface reconstructions. Splatter Image [50] merges Gaussian primitives from single-view regressions but lacks cross-view information, limiting its multiview applications. GPS-Gaussian [71] and MVSplat [9] improve feature matching with cost volumes for better geometries; however, GPS-Gaussian is limited to human body reconstruction with depth ground truth, and MVSplat, using plane-sweep stereo [28, 29, 62, 63], still relies on the epipolar priors [13, 15, 63]. Triplane-Gaussian [73] encodes single-view images into latent 3D point clouds and triplane features, outputting 3D Gaussian properties via MLP decoders. However, it focuses on single-view reconstruction, with rendering quality dependent on initial geometry. Our method bypasses 3D priors through sampling along epipolar lines or cost volumes, instead using cross-view competition pre-training [59, 60] on large-scale datasets [25, 35, 40, 44, 45].

**Solving 3D Tasks using Geometry-free Methods.** Priors are crucial for visual tasks to provide generalized features [14, 30, 31, 32, 33]. Capitalizing on the geometric priors, methods based on re-projection features [21, 51, 64], cost volume [7, 19, 22, 63], and image warping [5] have performed well in downstream 3D activities. However, these methods rely on task-specific designs and struggle with complex scenarios, such as occlusions or non-overlapping views. Recently, some geometry-free alternatives have been proposed to this challenge. SRT [43] and GS-LRM [68] are epipolar-free GNVS methods that boldly eschew any explicit geometric inductive biases. SRT encodes patches from all reference views using a Transformer encoder and decodes the RGB color for target rays through a Transformer decoder. GS-LRM's network, composed of a large number of Transformer blocks, implicitly learns 3D representations. However, due to the lack of targeted scene encoding, these methods are either limited to specific datasets or suffer from unacceptable computational efficiency and carbon footprint. Some pose-free GNVS methods [20, 43, 54] are also epipolar-free. These methods, lacking known camera poses, find it challenging to perform epipolar line sampling. They often reduce task complexity through specially designed feature representations (e.g., Learned 3D Neural Volume in LEAP [20] and Triplane in PF-LRM [54]), but this reduction comes at the cost of decreased model generalization. Different from the above methods, our method focuses on data-driven 3D priors and does not require any time-consuming and complex structured feature representations, such as cost volumes. CroCo [59], a self-supervised pre-training method for 3D vision tasks, uses cross-view completion to recover occluded parts of an image from different viewpoints without any 3D inductive biases, significantly enhancing downstream 3D vision tasks. DUSt3R [56] introduces a novel paradigm for dense and unconstrained stereo 3D reconstruction from arbitrary image collections, operating without prior information about camera calibration. These geometry-free pioneers pave the way for more adaptable and efficient 3D vision systems capable of performing accurately across diverse and challenging environments.

## 3 Methodology

### 3.1 Overview

Our objective is to predict per-pixel 3D Gaussian [23] primitives $\{\boldsymbol{\mu}_i, \boldsymbol{\Sigma}_i, \alpha_i, \boldsymbol{SH}_i\}_{i=1}^{M}$ using $N$ reference views images $\{\boldsymbol{I}_j\}_{j=1}^{N}$, camera intrinsics matrices $\{\boldsymbol{K}_j\}_{j=1}^{N}$ and poses matrices $\{\boldsymbol{P}_j\}_{j=1}^{N}$

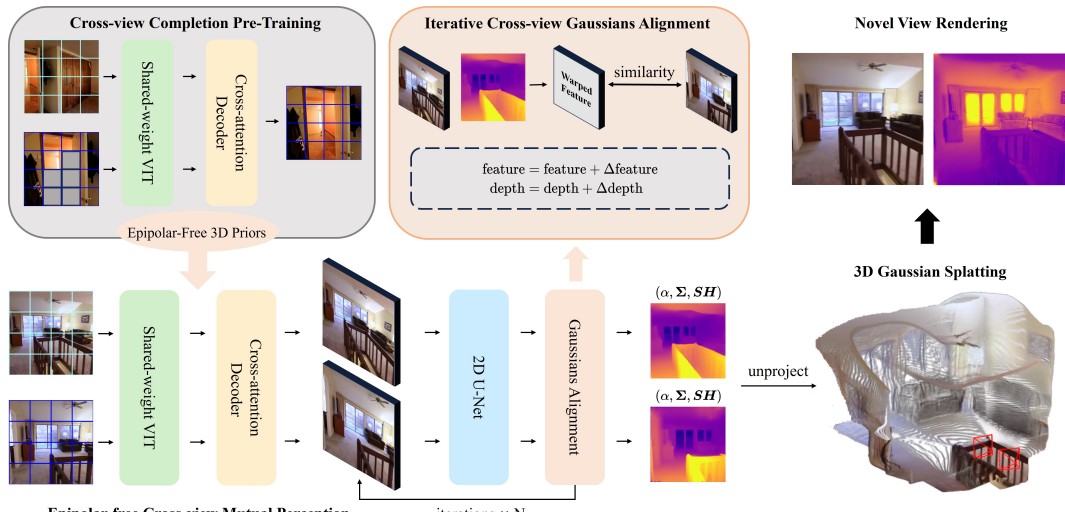

**Fig. 2.** Overview of eFreeSplat. (a) Epipolar-free Cross-view Mutual Perception leverages self-supervised cross-view completion pre-training [60] to extract robust 3D priors. The ViT [11] with shared weights processes the reference images, followed by a cross-attention decoder to generate multiview feature maps, forming 3D perception without epipolar priors. (b) Iterative Cross-view Gaussians Alignment module iteratively refines Gaussian attributes through a 2D U-Net. The process involves warped features to align corresponding features and depths, ensuring consistent depth scales across different views. (c) The final step involves employing rasterization-based volume rendering [23] to generate high-quality geometry and realistic novel view images.

in a single feed-forward inference. The 3D Gaussian primitives include position $\boldsymbol{\mu}$, covariance $\boldsymbol{\Sigma}$, opacity $\alpha$, and spherical harmonics $\boldsymbol{SH}$ for colors. Given a $H \times W$ sized reference image, the number of 3D Gaussian primitives can be calculated as $M = N \times H \times W$. The position of the 3D Gaussians $\boldsymbol{\mu}_i$ determines the geometric shape of the scene, which corresponds to pixel $\mathbf{u}$ is calculated using the camera origin $\boldsymbol{o}$, the ray direction $\boldsymbol{d}_{\mathbf{u}}$, and the predicted depth $d$:

$$\boldsymbol{\mu}_i = \boldsymbol{o} + d \cdot \boldsymbol{d}_{\mathbf{u}}, \tag{1}$$

where $\boldsymbol{d}_u$ is calculated by the camera intrinsic and pose matrix: $\boldsymbol{d}_{\mathbf{u}} = \boldsymbol{P}\boldsymbol{K}^{-1}[\mathbf{u}, 1]^T$. However, when the number of reference views is extremely sparse, predicting accurate depths $d$ and reconstructing high-quality geometric structures and appearances become particularly challenging. Particularly in non-overlapping and occluded areas, prevalent methods [6, 9, 12] based on epipolar line sampling fail to introduce valid geometric priors.

In this paper, we propose eFreeSplat, a generalizable 3D Gaussian Splatting model from sparse reference views[2] that operates independently of epipolar line priors. As illustrated in Fig. 2, the pre-trained ViT model based on cross-view completion via self-supervised training [59, 60] in large-scale datasets provides robust geometric priors, serving as our Epipolar-free Cross-view Mutual Perception (Sec. 3.2). Unlike recent works [6, 9, 61], which directly combine per-view 3D Gaussians, we propose Iterative Cross-view Gaussians Alignment (ICGA) in Sec. 3.3. This module iteratively updates the position and features of Gaussians by calculating the similarity between warped features and corresponding features, alleviating the issues of local geometric inaccuracies caused by inconsistent depth scales. In Sec. 3.4, we predict the centers of the 3D Gaussians by unprojecting the aligned depth maps while calculating other 3D Gaussian parameters based on the aligned features.

### 3.2 Epipolar-free Cross-view Mutual Perception

To realize the cross-view mutual preception without relying on the epipolar prior, we extract cross-view image features using a shared-weight ViT $\mathcal{E}_{\boldsymbol{\theta}_1}$ and a cross-attention decoder $\mathcal{D}_{\boldsymbol{\theta}_2}$, both pre-trained on large-scale cross-view completion tasks in a self-supervised manner [60]. Following

---

[2]In our experiments, the number of reference views $N = 2$, which is consistent with previous methods [6, 9]. For convenience, all subsequent discussions will assume a 2-views input scenario.

the methodologies of CroCo v2 [60] and ViT [11], both images $\boldsymbol{I}_1$ and $\boldsymbol{I}_2$ are divided into $2n$ non-overlapping patches via a linear projection, with each patch measuring $16 \times 16$ pixels. Additionally, relative positional embeddings [47] are added to the RGB patches before inputted into a series of stacked Transformer modules for encoding tokens $\varepsilon_j$:

$$\{\varepsilon_j\}_{j=1}^2 = \{\mathcal{E}_{\boldsymbol{\theta_1}}(\boldsymbol{I}_j)\}_{j=1}^2. \tag{2}$$

After encoding $\boldsymbol{I}_1$ and $\boldsymbol{I}_2$ via ViT independently, the cross-attention decoder $\mathcal{D}_{\boldsymbol{\theta_2}}$ takes $\varepsilon_1$ and $\varepsilon_2$ conditioned on each other for cross-view features $\boldsymbol{\mathcal{F}}_j \in \mathbb{R}^{C \times H \times W}$:

$$\boldsymbol{\mathcal{F}}_1 = f\left(\mathcal{D}_{\boldsymbol{\theta_2}}\left(\varepsilon_1, \varepsilon_2\right)\right), \quad \boldsymbol{\mathcal{F}}_2 = f\left(\mathcal{D}_{\boldsymbol{\theta_2}}\left(\varepsilon_2, \varepsilon_1\right)\right). \tag{3}$$

The structure of the cross-attention decoder consists of alternating multi-head self-attention blocks and multi-head cross-attention blocks. The mapping function $f$ refers to unflattening the tokens back to the original image size. The multi-head self-attention blocks learn token representations from the first viewpoint, while the multi-head cross-attention blocks facilitate cross-view information exchange conditioned on the token representations from the second view.

The CroCo model [59, 60], as a variant of masked image modeling [1, 16, 58] that leverages cross-view information from the same scene to capture the spatial relationship between two images, can significantly enhance performance on 3D downstream tasks. Based on cross-view completion self-supervised pre-training on large-scale datasets, our epipolar-Free cross-view mutual perception method provides robust 3D priors information by understanding the spatial relationship between the two images [59]. Due to the randomness of the masking process during pre-training, the pre-trained model is capable of reasoning about non-overlapping and occluded areas, which is hard for traditional geometric methods to achieve. Therefore, our epipolar-Free mutual perception possesses a more global and robust feature-matching inductive bias compared to methods [6, 9, 12, 61] that rely on epipolar line sampling [17] or the plane-sweep stereo approach [63].

### 3.3 Iterative Cross-view Gaussians Alignment

To address the issue of inconsistent depth scales across different views, we utilize cross-view feature matching information to align and update per-pixel Gaussians' centers and features iteratively.

Firstly, we predict per-pixel Gaussians' depths $\boldsymbol{d}$ and features $\boldsymbol{\mathcal{G}}$ via a 2D U-Net [42] mapping $U$ with cross-view attention, similar to [9]:

$$\boldsymbol{d}_1, \boldsymbol{\mathcal{G}}_1, \boldsymbol{d}_2, \boldsymbol{\mathcal{G}}_2 = U(\boldsymbol{\mathcal{F}}_1, \boldsymbol{\mathcal{F}}_2). \tag{4}$$

Next, to establish cross-view correspondences, we endeavor to make the features of each 3D Gaussian point projected onto the known camera planes to be as similar as possible. Taking the first view as an example, we calculate the warped features $\boldsymbol{\mathcal{G}}_{1,2}$ of the first view on the second view's features map via the predicted coarse depth $\boldsymbol{d}_1$:

$$\mathcal{W}_{1,2} = \boldsymbol{K}_2 \boldsymbol{R}_2 \left( \boldsymbol{R}_1^{-1} - \frac{\left(\boldsymbol{R}_2^{-1}\mathbf{t}_2 - \boldsymbol{R}_1^{-1}\mathbf{t}_1\right) \mathbf{n}_1^T}{\boldsymbol{d}_1} \right) \boldsymbol{K}_1^{-1}, \tag{5}$$

$$\boldsymbol{\mathcal{G}}_{1,2}(\mathbf{u}) = \boldsymbol{\mathcal{G}}_2(\mathcal{W}_{1,2}[\mathbf{u}, 1]^T), \tag{6}$$

where $\mathcal{W}$ denotes the homographic warping matrix. $\mathbf{u}$ represents a pixel location in the first view. $\boldsymbol{R}_i$ and $\boldsymbol{t}_i$ are the rotation and translation parameters of the camera pose $\boldsymbol{P}_i$. $\mathbf{n}_i$ refers to the normal vector of the target plane. We compute the similarity $\boldsymbol{\mathcal{S}}^1, \boldsymbol{\mathcal{S}}^2$ between the warped feature map $\boldsymbol{\mathcal{G}}_{1,2}$ and the corresponding feature map $\boldsymbol{\mathcal{G}}_1$ based on $\Delta\boldsymbol{d}_{\mathrm{cos}}$. $\boldsymbol{\mathcal{S}}^2$ is obtained by the dot product of $\boldsymbol{\mathcal{G}}_1$ and $\boldsymbol{\mathcal{G}}_{1,2}$, where $C$ denotes the feature dimension of the 3D Gaussian primitives.

$$\boldsymbol{\mathcal{S}}^1 = (\boldsymbol{\mathcal{G}}_1 - \boldsymbol{\mathcal{G}}_{1,2})^2, \quad \boldsymbol{\mathcal{S}}^2 = \frac{\boldsymbol{\mathcal{G}}_1 \cdot \boldsymbol{\mathcal{G}}_{1,2}}{\sqrt{C}}. \tag{7}$$

Finally, we update the coarse per-pixel 3D Gaussian features and predicted depths.

$$\Delta\boldsymbol{\mathcal{G}}_1 = \varphi([\,\boldsymbol{\mathcal{G}}_1 \parallel \boldsymbol{\mathcal{S}}^1\,]) \cdot \boldsymbol{\mathcal{S}}^2, \quad \Delta\boldsymbol{d}_1 = \boldsymbol{d}_1 \cdot \boldsymbol{\mathcal{S}}^2, \tag{8}$$

$$\boldsymbol{\mathcal{G}}_1 = \boldsymbol{\mathcal{G}}_1 + \Delta\boldsymbol{\mathcal{G}}_1, \quad \boldsymbol{d}_1 = \boldsymbol{d}_1 + \Delta\boldsymbol{d}_1, \tag{9}$$

**Table 1.** Quantitative comparisons. We evaluate our method by rendering three novel view images from two reference viewpoints for each scene. The performance is determined by averaging across all scenes. The dataset's training and testing split follows the protocol established by pixelSplat [6]. The inference time includes both scene encoding and rendering time, tested on a single RTX-4090 GPU.

| Methods | RealEstate10K [72] | | | ACID [26] | | | Inference Time |
| | PSNR↑ | SSIM↑ | LPIPS↓ | PSNR↑ | SSIM↑ | LPIPS↓ | (s) |
|---|---|---|---|---|---|---|---|
| Du et al. [12] | 24.78 | 0.820 | 0.213 | 26.88 | 0.799 | 0.218 | 1.578 |
| GPNR [49] | 24.11 | 0.793 | 0.255 | 25.28 | 0.764 | 0.332 | 13.180 |
| pixelSplat [6] | 25.89 | 0.858 | 0.142 | 28.14 | 0.839 | 0.150 | 0.100 |
| MVSplat [9] | 26.39 | **0.869** | 0.128 | 28.25 | 0.843 | 0.144 | **0.046** |
| eFreeSplat | **26.45** | 0.865 | **0.126** | **28.30** | 0.851 | **0.140** | 0.061 |

where $[\cdot\|\cdot]$ refers to the concatenation operation of tensors. We employ the mapping function $\varphi : \mathbb{R}^{2C \times H \times W} \mapsto \mathbb{R}^{C \times H \times W}$ through lightweight convolutional blocks.

The updated features and depths serve as inputs for Eq. (4) (5) and (6), bootstrapping the next iteration of Gaussian updates. Our cross-view Gaussians alignment method, during each iteration, involves establishing a match for target pixel $\mathbf{u}_1$ in the first view with matching pixel $\mathbf{u}_2$ in the second view. This process is akin to considering all neighboring pixels of the projected pixel $\mathbf{u}_2'$ based on the current coarse depth due to the locality inductive bias inherent in convolutions. During each querying process, the discrepancy between $\mathbf{u}_2'$ and the true matching $\mathbf{u}_2$ progressively decreases, thereby harmonizing the consistency of depth scales across multiple views.

### 3.4 Gaussian Parameters Prediction

We calculate the per-view Gaussians' centers $\boldsymbol{\mu}$ based on the refined depths and camera parameters using Eq. (1). We predict additional Gaussian primitives: $\boldsymbol{\Sigma}, \alpha, \boldsymbol{SH}$ , via an additional U-Net. Following other 3DGS-based methods [6, 9, 23], the covariance matrix $\boldsymbol{\Sigma}$ is composed of a scaling matrix and a rotation matrix. The spherical harmonic coefficients $\boldsymbol{SH}$ are used to compute RGB values given a direction. Since we have harmonized the depth scale across different viewpoints, we directly merge all views' Gaussian primitives $\{\boldsymbol{\mu}_i, \boldsymbol{\Sigma}_i, \alpha_i, \boldsymbol{SH}_i\}_{i=1}^{N \times H \times W}$.

## 4 Experiments

### 4.1 Experimental Settings

**Datasets.** eFreeSplat is trained on RealEstate10K [72] and ACID [26]. The RealEstate10K dataset consists of home tour videos, providing a wealth of scenes and a variety of viewpoint changes. The ACID dataset contains aerial landscape videos, featuring expansive views and complex terrains. Both datasets provide estimated camera parameters. Following pixelSplat [6], we use the provided training and testing splits and evaluate three novel view images on each test scene.

**Evaluation Metrics and Training Losses.** We employ standard image quality metrics to validate and compare our results quantitatively: pixel-level PSNR, patch-level SSIM [57], and feature-level LPIPS [69]. During the training phase, the loss is composed of a linear combination of MSE and LPIPS loss, with loss weights of 1 and 0.05, respectively. Since existing methods conduct experiments at $256 \times 256$, we also set the resolution of our training and testing images for fair comparison.

**Comparison Methods.** We compared four feed-forward methods for sparse view novel view synthesis. Du et al. [12] and GPNR [49] are the methods based on light field rendering that combines features on epipolar lines aggregated by the epipolar transformer. pixelSplat [6] and MVSplat [9] are the latest 3DGS-based models based on epipolar sampling and multi-plane sweeping, respectively. Our method compared the qualitative and quantitative results with these four methods.

**Implementation details.** The ViT-B vision transformer [11] and cross-attention decoder [59] have been pretrained by CroCo v2 [60], which underwent self-supervised cross-view completion training on large-scale datasets [25, 35, 40, 44, 45]. The Iterative Alignment and Updating strategy

| Ref. | Du et al. [12] | pixelSplat [6] | MVSplat [9] | eFreeSplat | Ground Truth |
|---|---|---|---|---|---|

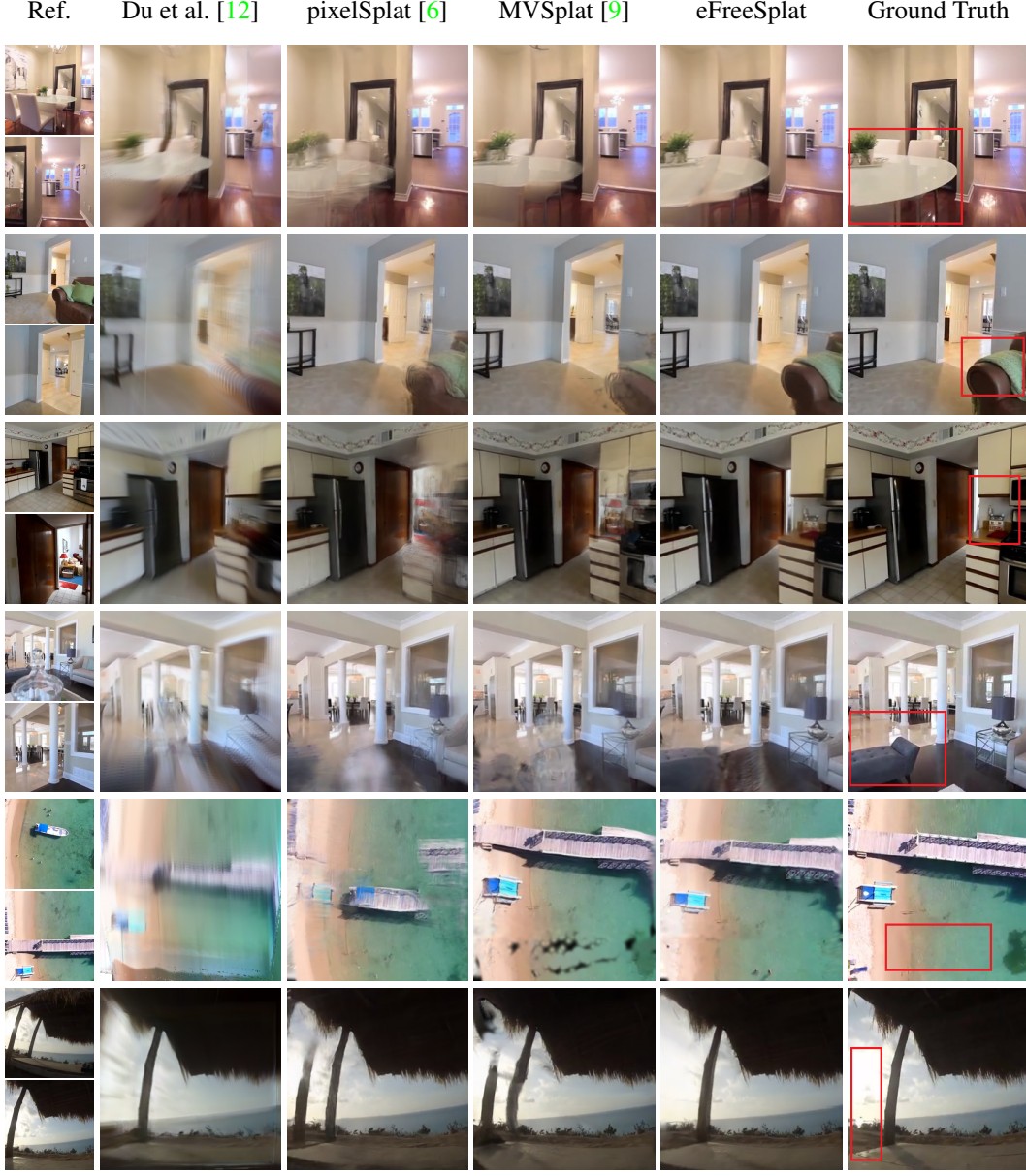

**Fig. 3.** We provide qualitative comparisons on the RealEstate10K (first four rows) and the ACID (last two rows). Compared to baselines, our method produces fewer artifacts in rendering results (red boxes). Moreover, our approach can perform better in non-overlapping areas (1st, 2nd, 5th and 6th rows) and occluded areas ( 3th and 4th rows) without relying on unreliable epipolar priors.

is implemented through 2 iterations. All models are trained on 4 RTX-4090 GPUs for $300,000$ iterations using the Adam optimizer [24]. More details are provided in Appendix C.

## 4.2  Comparative Studies

**Image quality comparison.** We report quantitative results against baselines [6, 9, 12, 49] on the RealEstate10K and ACID datasets in Tab. 1. Our method, eFreeSplat, outperforms the SOTA method, MVSplat [9] by 0.06dB in PSNR on the RealEstate10K dataset and by 0.05dB on the ACID dataset. The evaluation metrics for all baselines are derived from experimental results published in the papers on pixelSplat [6] and MVSplat [9].

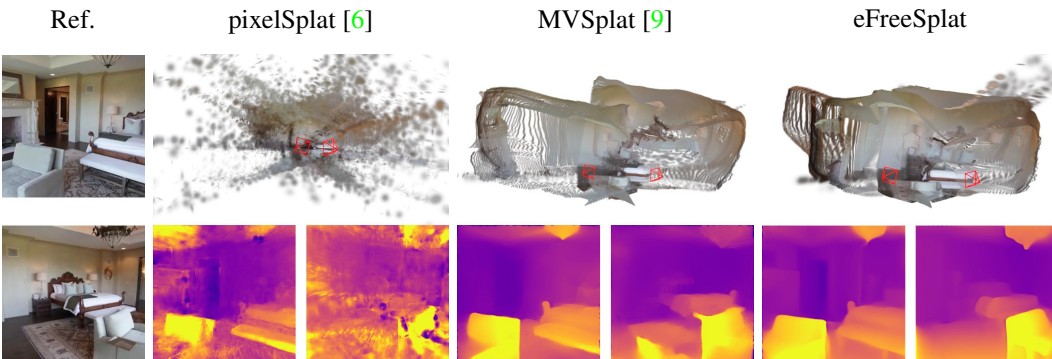

| Ref. | pixelSplat [6] | MVSplat [9] | eFreeSplat |

**Fig. 4.** Comparison results about 3D Gaussians (top) and predicted depth maps of the reference viewpoints (bottom). Compared to SOTA 3DGS-based methods [6, 9], our method achieves higher quality in 3D Gaussian Splatting and produces smoother depth maps.

**Table 2.** In more challenging scenarios, we classify the RealEstate10K dataset [72] into three subsets based on the overlap size of the reference images: scenes with an overlap below 0.7, 0.6, and 0.5.

| Methods | Overlap 0.7 | | | Overlap 0.6 | | | Overlap 0.5 | | |
| | PSNR↑ | SSIM↑ | LPIPS↓ | PSNR↑ | SSIM↑ | LPIPS↓ | PSNR↑ | SSIM↑ | LPIPS↓ |
|---|---|---|---|---|---|---|---|---|---|
| pixelSplat [6] | 25.05 | 0.852 | 0.145 | _24.79_ | _0.849_ | 0.149 | _24.96_ | _0.846_ | _0.149_ |
| MVSplat [9] | _25.11_ | _0.854_ | _0.139_ | 24.70 | 0.841 | _0.146_ | 24.64 | 0.840 | 0.150 |
| eFreeSplat | **25.72** | **0.861** | **0.132** | **25.48** | **0.859** | **0.135** | **25.46** | **0.853** | **0.139** |

Our method's qualitative comparison with baselines is illustrated in Fig. 3. Our rendering results show fewer artifacts or object deformations, especially in non-overlapping or occluded areas. Competitive methods like pixelSplat [64] and MVSplat [9], based on sampling along the epipolar lines, produce unreliable reconstructions in these challenging areas. It demonstrates that eFreeSplat provides more robust 3D priors than epipolar priors, offering global 3D perception even in challenging areas.

**Geometry quality comparison.** As illustrated in Fig. 4, our method produces higher-quality 3DGS reconstructions and smoother priors without the epipolar priors. pixel-Splat [6], despite additional finetuning via depth regularization during training, exhibits noticeable artifacts in its reconstructed 3DGS and depth maps. MVSplat [9] generates competitive depth maps by building a cost volume representation [63], which directly merges per-view Gaussians, resulting in significant point cloud shifts. Our method, which does not rely on sampling along epipolar lines or additional depth regularization finetuning, surpasses current SOTA methods in 3DGS reconstruction quality. Please refer to Appendix A for additional comparison and analysis.

**Performance with Low-overlapped observations.** In this section, we analyze the differences between our method and 3DGS-based methods when the reference viewpoints have a lower overlap. First, we

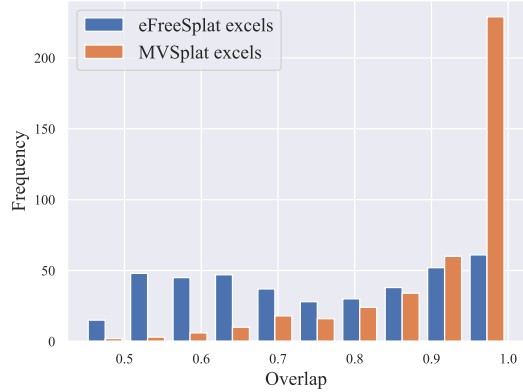

**Fig. 5.** Our method reconstructs more reliable results than MVSplat when the reference views overlap is low. In the histogram, the blue bars represent the frequency at which our method exceeds MVSplat in rendering quality under the current overlap conditions, while the orange bars indicate the opposite.

counted the number of scenes where our method and MVSplat [9] outperform each other in PSNR on the RealEstate10K dataset, selecting the top 400 scenes with the largest PSNR differences for

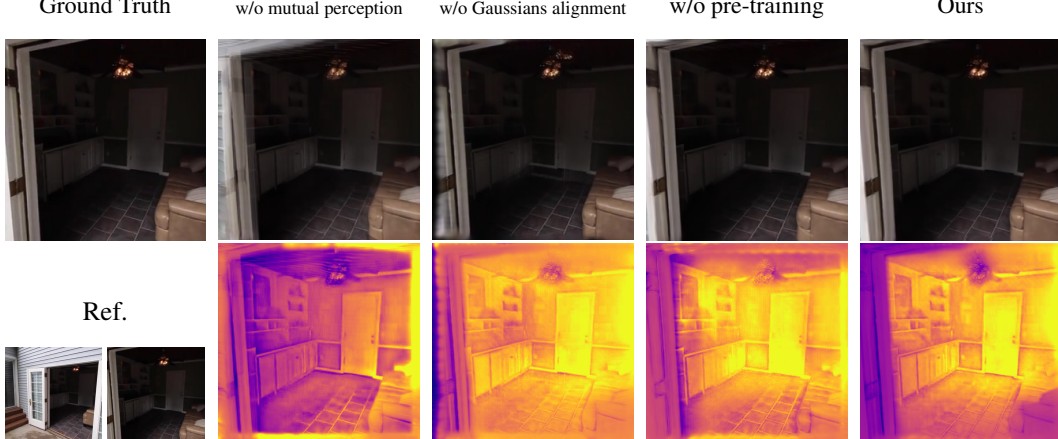

| Ground Truth | w/o mutual perception | w/o Gaussians alignment | w/o pre-training | Ours |

**Fig. 6.** Ablations. The first row displays the novel viewpoint images, while the last row shows the reference viewpoints and the depth maps of the novel views. Our full model renders higher-quality RGB images and smoother depth maps.

**Table 3.** Ablations. All ablation experiments were conducted by training and evaluating on the RealEstate10K dataset [72]. Each ablation model was derived from our full model by removing the corresponding modules.

| Model | PSNR↑ | SSIM↑ | LPIPS↓ |
|---|---|---|---|
| eFreeSplat (Full) | **26.45** | **0.865** | **0.126** |
| w/o mutual perception | 22.04 | 0.723 | 0.212 |
| w/o Gaussians alignment | 23.03 | 0.758 | 0.187 |
| w/o pre-training weights | 24.81 | 0.829 | 0.153 |

each. As shown in Fig. 5, our method performs better in scenes with more minor viewpoint overlaps, while MVSplat excels when the overlap is close to 1. In Tab. 2, our method outperforms other 3DGS baselines [6, 9] in settings with more minor overlaps by 3.1% ↑ in PSNR and 8.6% ↓ in LPIPS. It confirms the robustness of our method in non-overlapping areas. However, methods based on epipolar priors have advantages in scenes where reference viewpoints are closer, and reconstruction quality declines as the overlap decreases.

### 4.3 Ablation Studies

As shown in Tab. 3 and Fig. 6, we conducted ablation studies on the eFreeSplat model on the RealEstate10K dataset. We will detail the analysis in the following three subsections.

**Importance of epipolar-free cross-view mutual perception.** Epipolar-free cross-view mutual perception extracts cross-view image features using a shared-weight ViT [11] and a cross-attention decoder. According to Tab. 3, this module's absence results in a 4.41dB decrease in PSNR. In Fig. 6, the absence of cross-view mutual perception results in significant offsets in the depth map and noticeable artifacts.

**Importance of iterative cross-view Gaussians alignment.** Iterative cross-view Gaussian alignment updates per-pixel Gaussian features and depths through warped U-Net features, thereby aligning the cross-view 3D Gaussian point clouds. The lack of Gaussian alignment can lead to pixel displacement or unreliable local geometric details (*e.g.*, the lamp's position in Fig 6). Additionally, we conducted extra experiments with 1 to 3 iterations. As shown in Fig. 7, using 2 iterations significantly reduces artifacts and inconsistent depth in novel view rendering. This validates that the iterative mode helps align the depth scale across multiple views. When the iteration count increases to 3, there is no notable improvement in reconstruction and rendering quality. For further analysis and results, please refer to Appendix B.

**Importance of self-supervised cross-view completion pre-training.** In Fig. 6, the absence of cross-view completion pre-training weights results in unaccuracy depth maps. Self-supervised pre-

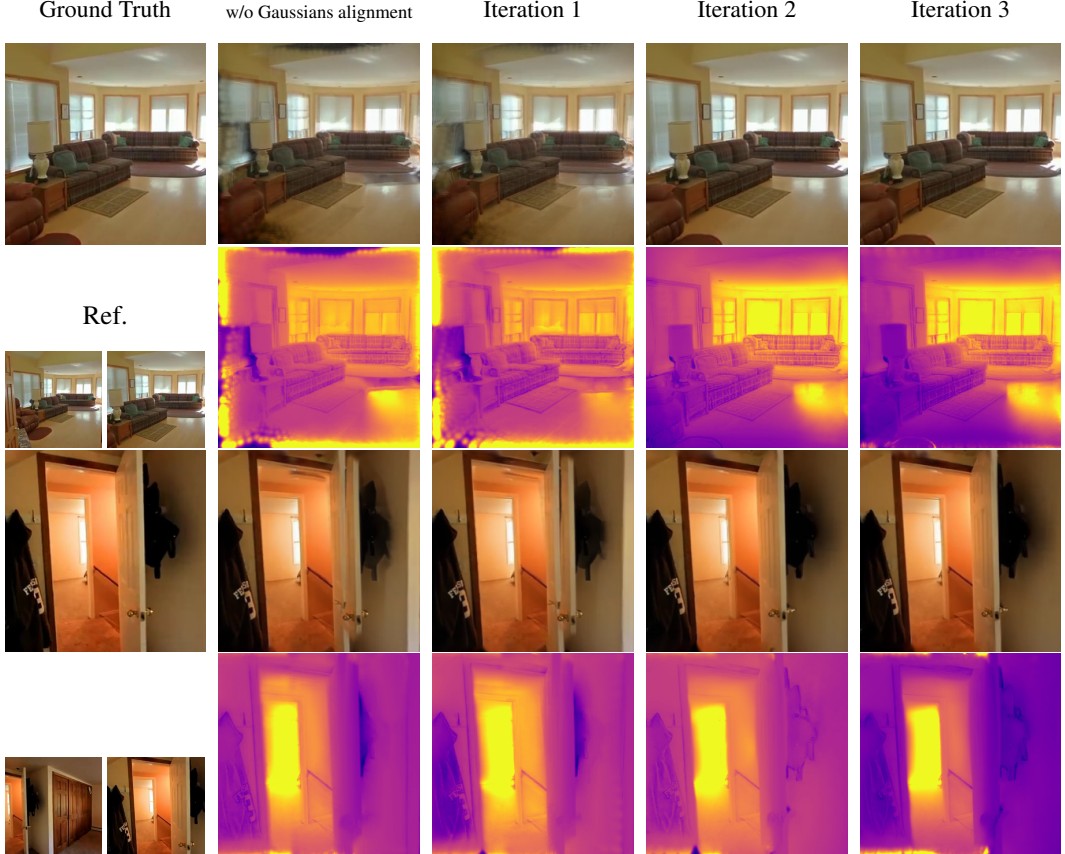

**Fig. 7.** Ablation of gaussians alignment module. Additional iterations can significantly aid in aligning the depth scale and reducing artifacts that occur during novel view rendering.

training by cross-view completion [60] on large-scale datasets allows our model to perceive spatial correspondences, thereby enabling it to predict more reliable and smoother depth maps.

## 5   Conclusion

Our work introduces eFreeSplat, a novel generalizable 3D Gaussian Splatting model tailored for novel view synthesis across new scenes, designed to function independently of epipolar constraints that might be unreliable when large viewpoint changes occur. By leveraging a Vision Transformer architecture self-supervised pre-trained by cross-view completion [60] on large-scale datasets, eFreeSplat excels in handling sparse and challenging viewing conditions that traditional methods [17, 63] struggle with. This model's ability to unify the consistency of depth scales across different views marks a significant improvement over existing techniques, effectively addressing issues like artifacts and misalignment in rendered images. Our experiments have demonstrated that our method provides high-quality geometric reconstructions and novel viewpoint images. In settings with a large baseline from 2-view inputs, it outperforms the latest state-of-the-art methods [6, 9] that rely on epipolar priors.

## 6   Acknowledgement

This work was supported by the National Natural Science Foundation of China (62293554, 62206249, U2336212), "Pioneer" and "Leading Goose" R&D Program of Zhejiang (2024C01073), Ningbo Innovation "Yongjiang 2035" Key Research and Development Programme (2024Z292).

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

# A   Additional Experimental results

We provide additional qualitative comparisons against baselines. The visualization results on the RealEstate10K are shown in Fig. 8. Additionally, we provide more geometry reconstruction comparison results, as shown in Fig. 9. Our method reconstructs high-quality 3DGS without using epipolar priors or depth regularization finetuning.

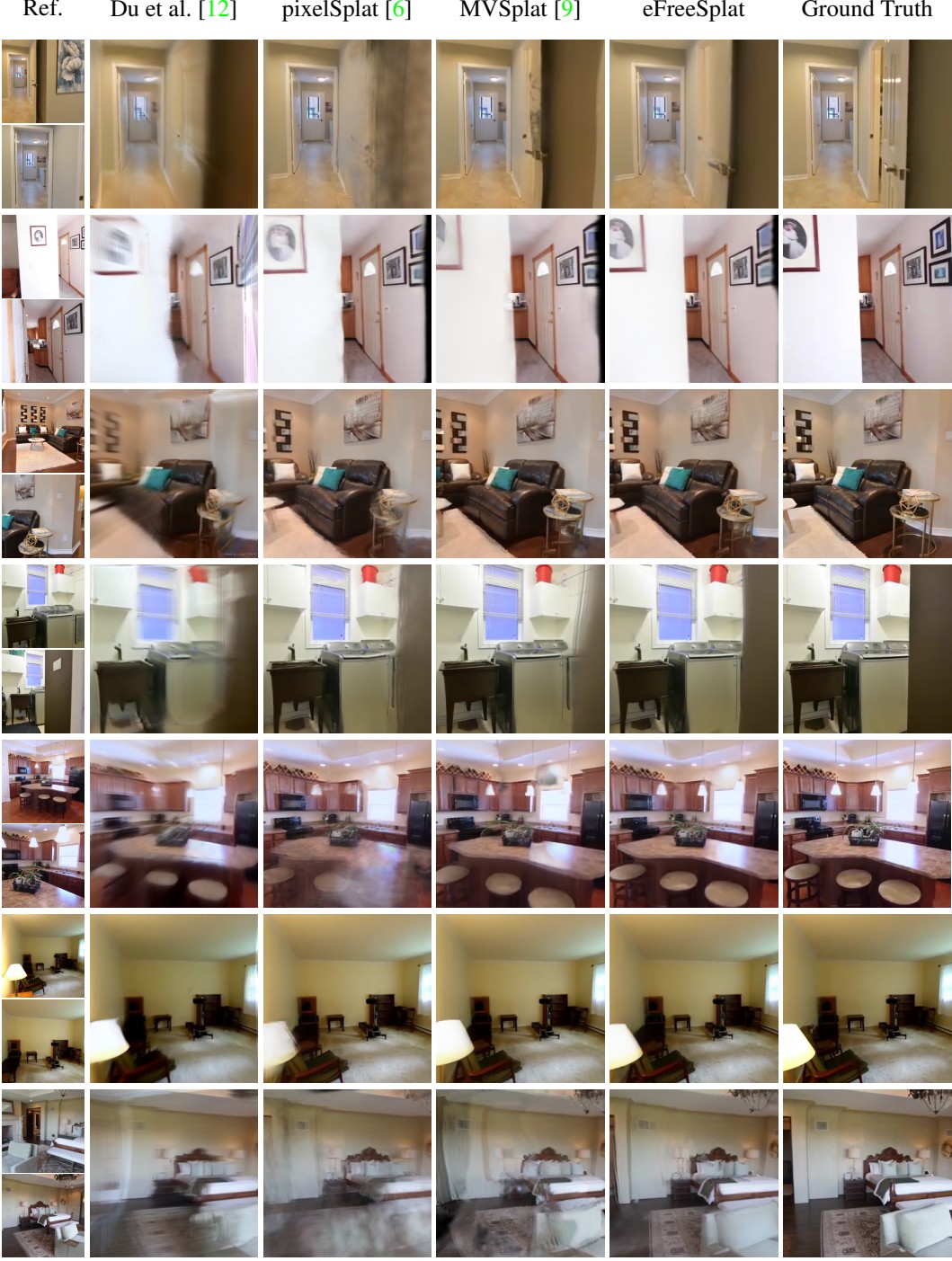

| Ref. | Du et al. [12] | pixelSplat [6] | MVSplat [9] | eFreeSplat | Ground Truth |

**Fig. 8.** Additional visualization results on the RealEstate10K [72]. Our method, eFreeSplat, outperforms baselines in rendering results, producing fewer artifacts and scene distortions.

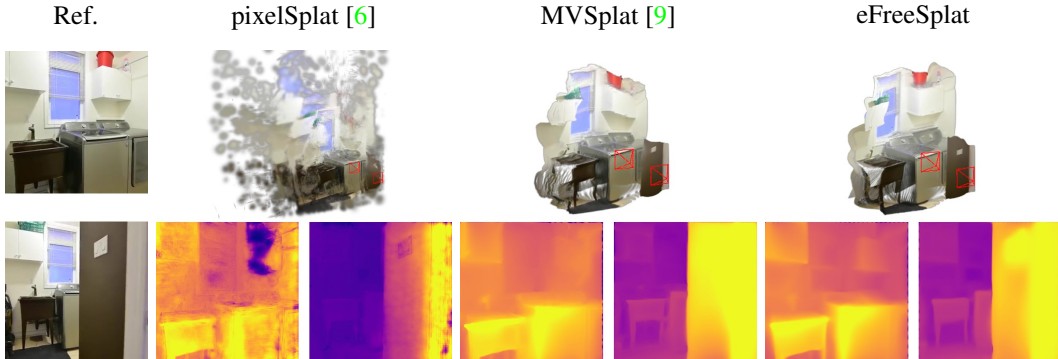

| Ref. | pixelSplat [6] | MVSplat [9] | eFreeSplat |
|------|----------------|-------------|------------|

**Fig. 9.** Additional geometry reconstruction quality comparison results. Our method achieves higher quality in 3D Gaussian Splatting and produces smoother depth maps than pixelSplat [6] and MVSplat [9].

**Table 4.** Quantitative results of gaussians alignment module under the settings of 1 to 3. "Memory" refers to GPU memory usage, and "Time" indicates the inference time.

| Iterations | PSNR↑ | SSIM↑ | LPIPS↓ | Memory(M) | Times(s) |
|------------|-------|-------|--------|-----------|----------|
| 1 | 23.36 | 0.768 | 0.182 | 2410 | 0.058 |
| 2 | **26.45** | **0.865** | 0.126 | 2452 | 0.061 |
| 3 | 26.40 | 0.861 | **0.126** | 2488 | 0.086 |

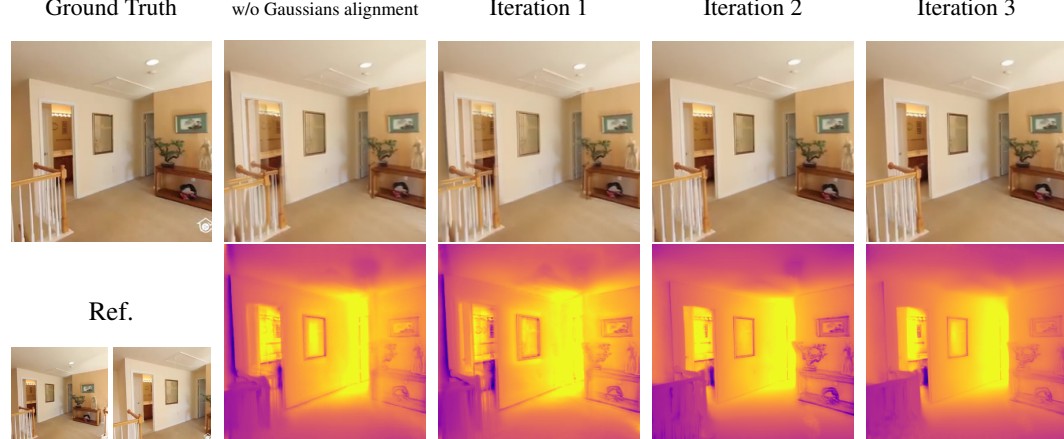

| Ground Truth | w/o Gaussians alignment | Iteration 1 | Iteration 2 | Iteration 3 |
|--------------|------------------------|-------------|-------------|-------------|

Ref.

**Fig. 10.** Visualization of gaussians alignment module under the settings of 1 to 3.

## B Additional Experimental Analysis

**More ablations.** In this section, we provide both quantitative and qualitative results of the Gaussians Alignment module under the settings of 1 to 3 iterations. As shown in Tab. 4 and Fig. 10, setting the iteration count to 2 effectively reduces artifacts caused by inconsistent depth scales. When the iteration count is set to 3, we had to reduce the model's parameter size and batch size to avoid OOM errors, which might be one of the reasons for the lack of significant improvement in image reconstruction metrics.

**Failure cases.** Our method relies on the 3D prior knowledge provided by CroCo [59] pre-trained weights. However, the input viewpoint overlap in the pre-trained dataset does not exceed 0.75 [60], while the input viewpoint overlap in the RealEstate10K and ACID datasets mainly ranges from 0.9 to 1.0. As shown in Fig. 11, our method renders unreliable results when the input viewpoints are very close, which can be attributed to the distribution bias between the GNVS dataset [26, 72] and the pre-trained dataset [25, 35, 40, 44, 45].

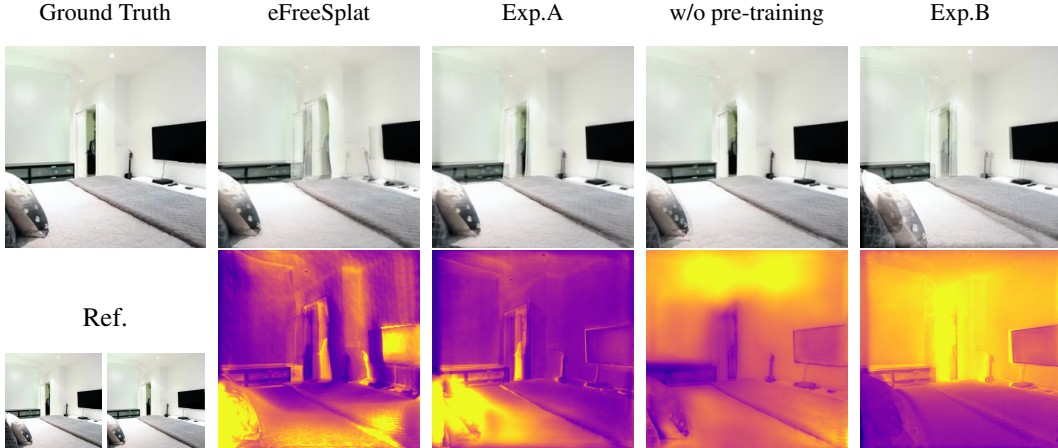

| Ground Truth | eFreeSplat | Exp.A | w/o pre-training | Exp.B |

Ref.

**Fig. 11.** Failure cases. Our method may produce unstable results in scenarios where the input viewpoints are very close. Exp.A and Exp.B indicate that fine-tuning the CroCo model on Re10k helps mitigate this issue.

**Table 5.** Exp. A involves fine-tuning the CroCo pretrained weights using the RE10K training set, while Exp. B trains CroCo directly using the RE10K training set without loading the pretrained weights. w/o pre-training refers to neither using CroCo pre-trained weights nor performing fine-tuning.

| Model | PSNR↑ | SSIM↑ | LPIPS↓ |
|---|---|---|---|
| raw eFreeSplat | **26.45** | **0.865** | **0.126** |
| Exp.A | 26.32 | 0.862 | 0.129 |
| w/o pre-training | 24.81 | 0.829 | 0.153 |
| Exp.B | **25.12** | **0.839** | **0.144** |

**Limitations.** Our method lacks geometric inductive biases, so our model is data-hungry and sensitive to the training data distribution. Joint training with richer multiview datasets across different scenes could be a viable direction. Additionally, the per-pixel 3D Gaussian mapping struggles to reconstruct parts of the scene that are occluded or missing from input viewpoints, such as an obscured chair. Therefore, introducing high-level features for scene completion might be a future research direction for generalizable 3D Gaussian Splatting work.

**Fine-tuning of the CroCo model.** We have conducted preliminary explorations to address the aforementioned limitation. We conducted relevant Experiments A and B regarding fine-tuning the CroCo model using the RE10K dataset. Experiment A involved fine-tuning the CroCo pretrained weights with the RE10K training set, while Experiment B involved training CroCo directly with the RE10K training set without loading the pretrained weights. Finally, we retrained eFreeSplat using the new pretrained weights. As shown in Fig. 11 and Tab. 5, the results indicate that pretraining the backbone model on the RE10K training set effectively addresses the model's poor performance in low-overlap scenarios. However, in the RE10K test set, Experiment A's reconstruction metrics were slightly lower than those of the original model, which may be due to insufficient training iterations. We will further investigate the positive impact of fine-tuning the CroCo pretrained model on novel view synthesis and 3D reconstruction in future work.

**Potential negative societal impacts.** Our model could be misused for unethical purposes, such as creating false evidence or manipulating media, which threatens information integrity and personal privacy. Additionally, the model introduces security risks in contexts like autonomous driving, as it may produce incorrect reconstructions in real and complex scenarios. These concerns underscore the importance of implementing stringent ethical guidelines and security measures when deploying such technology, to prevent misuse and ensure that it is used responsibly.

# C  Additional Implementation Details

**The cross-attention decoder.**  Following the *CrossBlock* decoder architecture in CroCo [59], the cross-attention decoder comprises a self-attention module and a cross-attention module. Let $\varepsilon_1, \varepsilon_2 \in \mathbb{R}^{N \times C}$ be the tokens of the two viewpoints outputted by a Vision Transformer [11]. The computation process of the decoder is as follows:

$$
\begin{aligned}
\bar{\varepsilon}_i &= \text{ LayerNorm } (\varepsilon_i), & i &= 1, 2 \\
\varepsilon'_i &= \varepsilon_i + \text{Attention} \left( \bar{\varepsilon}_i, \bar{\varepsilon}_i, \bar{\varepsilon}_i \right), & i &= 1, 2 \\
\varepsilon''_1 &= \varepsilon'_1 + \text{Attention} \left( \text{LayerNorm} \left( \varepsilon'_1 \right), \bar{\varepsilon}_2, \bar{\varepsilon}_2 \right), & & (10) \\
\varepsilon''_2 &= \varepsilon'_2 + \text{Attention} \left( \text{LayerNorm} \left( \varepsilon'_2 \right), \bar{\varepsilon}_1, \bar{\varepsilon}_1 \right), & & \\
\text{output}_i &= \varepsilon''_i + \text{MLP} \left( \text{LayerNorm} \left( \varepsilon''_i \right) \right), & i &= 1, 2
\end{aligned}
$$

In Equations 10, $\text{Attention}$ is derived from the classic attention computation. The inputs $Q, K, V$ undergo projection transformations using $W_q, W_k, W_v$:

$$
Q' = W_q Q, \ K' = W_k K, \ V' = W_v V,
$$
$$
\text{Attention}(Q, K, V) = \text{Linear} \left( \text{softmax} \left( \frac{Q' K'^{\top}}{\sqrt{C}} \right) V' \right). \tag{11}
$$

**The cross-view U-Net.**  For the Gaussian Alignment Strategy and the prediction of Gaussian primitives, we utilize a 2D Cross-View U-Net inspired by [41, 52], 2024. We concatenate and flatten multiview feature maps for cross-view information exchange, similar to the structure of the U-Net used for cost volume refinement in MVSplat [9]. Specifically, for the Gaussian Alignment Strategy, we apply four times of $2 \times$ down-sampling and add attention at the $16 \times$ down-sampled level, with the channel dimensions being [32, 32, 64, 128, 256]. For the prediction of Gaussian primitives, we keep the channel dimension fixed at 32, while the rest of the architecture remains the same as that of the U-Net used in the Gaussian Alignment Strategy.

**More training details.**  Our model is trained and tested on 4 RTX-4090 GPUs using the Adam optimizer with a learning rate 2e-4. The per-GPU batch size during training is 4. Similar to pixelSplat [6], the distance between the two input viewpoints gradually increases throughout training. However, to learn more robust 3D prior information, our setup allows for a maximum viewpoint distance of 60 frames, compared to the 45 frames used by pixelSplat [6] and MVSplat [9].

