# OpenReview forum: "Epipolar-Free 3D Gaussian Splatting for Generalizable Novel View Synthesis"
_NeurIPS.cc/2024/Conference — NeurIPS 2024 poster_

### Official Review · Reviewer_1Kz3 · 2024-07-11

**Soundness:** 3
**Presentation:** 2
**Contribution:** 2
**Rating:** 5
**Confidence:** 5

**Summary:**

This paper tackles the generalizable NVS problem from sparse-view inputs. The main claim is that relying on epipolar geometry cues harms the performance, and, thus, it removes these geometric priors. It adopts the pre-training and supervised reconstruction/NVS training pipeline as in Dust3r. The authors verify the performance on RealEstate and ACID.

**Strengths:**

- The proposed method demonstrates comparable performance to the current SoTA method MVSplat.
- I like the idea of not using epipolar geometry priors, but I do have some problems regarding the claims and technique. Please see details in weaknesses.
- I like the idea of doing ablation with different input view overlap ratio, which is trying to convince me on their claim in the introduction and Figure 1.

**Weaknesses:**

- Overclaim. The authors claim that the proposed method is the first epipolar-free work (Line 64), but actually there are a lot prior works that are epipolar-free, espeicially for those works doing reconstruction from unposed images where the epipolar lines cannot be computed without the access of ground-truth poses. These works including SRT (the unposed version UpSRT), LEAP [1], PF-LRM [2] and Dust3r+Croco, where they do cross-view attention between different input views. LEAP also shows visualization results of the cross-view attention for implicit cross-view feature matching and correspondence. Besides, there are also some of lightfield-based methods are epipolar-free. I don't know why the authors have already discussed SRT and Dust3r+Croco in the related work section, but they still think they are the first epipolar-free work. All previously mentioned works should be discussed properly.

[1] Jiang, Hanwen et al. “LEAP: Liberate Sparse-view 3D Modeling from Camera Poses.” ICLR 2024.
[2] Wang, Peng et al. “PF-LRM: Pose-Free Large Reconstruction Model for Joint Pose and Shape Prediction.” ICLR 2024.

- Related work. More related work on generalizable NVS using gaussian splatting should be discussed, for example, GS-LRM [3].

[3] Zhang, Kai et al. “GS-LRM: Large Reconstruction Model for 3D Gaussian Splatting.” ECCV 2024.

- Evaluation. The evaluation doesn't verify the claims of the authors. Even though the idea of Table 2 is good, I don't see a clear different of the gain over the epipolar-aware baseline pixelSplat when the overlap is smaller.

**Questions:**

Please see the weaknesses.

**Limitations:**

The biggest concern is the novelty. This paper is a combination of Duster (cross-view pre-training + 2-view geometry without epipolar prior) and generalizable NVS using Gaussian Splatting (e.g. MVSplat).

I believe the authors need to discuss the prior works more properly and better word their contributions.

---

> ### Author Rebuttal · Authors · 2024-08-07
>
> ### **Weaknesses 1. Overclaim Issue**
> We would like to clarify any confusion regarding our use of the term "epipolar-free." This term specifically indicates our method's avoidance of epipolar sampling and cost volume techniques, which are commonly used in generalizable novel view synthesis (GNVS) tasks. While approaches like UpSRT, LEAP, and PF-LRM also do not utilize epipolar geometry, our method targets multiview GNVS tasks with precise camera poses. Thus, our approach is epipolar-free within the GNVS context but not entirely devoid of geometric information. As mentioned in the related work section, "Solving 3D Tasks using Geometry-free Methods," our focus is distinct from those methods that are entirely geometry-free, such as LEAP and PF-LRM, which do not directly apply to the GNVS tasks addressed in this paper.
> We will reiterate our method's contributions and intuition. The novelty of our approach lies in the fact that most current multi-view GNVS methods heavily rely on epipolar line sampling or cost volume, which struggle to provide effective priors in non-overlapping and occluded areas. Therefore, we innovatively utilize a 3D cross-view pretraining model to obtain epipolar-free 3D priors and the cross-view Gaussians Alignment module to acquire accurate Gaussian attribute-matching features. The GNVS experiments demonstrate the advantages of our epipolar-free approach over existing methods dependent on epipolar line sampling or cost volume techniques.
>
> ### **Weaknesses 2. Related Work**
> We appreciate your suggestion to include more related works on GNVS using Gaussian splatting. To the best of our ability, we have discussed 6 recent works on GNVS using Gaussian splatting, except for some studies very close to the submission date, such as GS-LRM. We will add discussions on GS-LRM and other relevant works to provide a more comprehensive context for our contributions.
>
> ### **Weaknesses 3. Evaluation**
> We appreciate your valuable feedback on our evaluation section. We would like to emphasize that our method demonstrates clear advantages over both pixelSplat and MVSplat when the overlap is smaller.
> Specifically, our method achieves an improvement of 0.5dB PSNR compared to pixelSplat, along with faster rendering speeds. This improvement is comparable to the PSNR gain over pixelSplat reported in the MVSplat paper on the RE10K dataset. According to the evaluation criteria in the MVSplat paper, a PSNR increase of 0.5dB is considered a relatively significant improvement.

---

> > ### Comment · Reviewer_1Kz3 · 2024-08-09
> >
> > Thanks for the reply. However, the rebuttal didn't solve my concerns, especially the over-claiming.
> >
> > The rebuttal says "While approaches like UpSRT, LEAP, and PF-LRM also do not utilize epipolar geometry, our method targets multiview GNVS tasks with precise camera poses." As far as I know, SRT can also do GNVS with perfect poses without using epipolar geometry. Besides, LEAP or PF-LRM can also do GNVS, and actually using GT poses in this work is a stronger assumption. I cannot agree that this paper is the first epipolar-free GNVS work.

---

> > > ### Author Response · Authors · 2024-08-12
> > >
> > > Thank you for your professional feedback. We believe our work does not involve any over-claiming.
> > >
> > > Firstly, "epipolar-free" and "geometry-free" are two distinct concepts. In our method, "epipolar-free" means that epipolar line sampling is not required, but it does not mean that we completely avoid using geometric information. In fact, our Iterative Cross-view Gaussians Alignment method relies on the warp transformation formula from the known geometry, as shown in Eq. (5) of the manuscript. In contrast, methods like SRT and GS-LRM are entirely geometry-free, meaning they do not use geometric information at all, making them true "geometry-free" methods. Our "epipolar-free" approach is more akin to a semi-geometry approach, so SRT and GS-LRM do not fall under the category of epipolar-free methods as proposed in this paper. We will further clarify the distinction between epipolar-free and geometry-free methods in the related works section of the revised version.
> > >
> > > Secondly, at this stage, unposed GNVS tasks and GNVS tasks with known poses are two different tasks. Here are three reasons to illustrate this:
> > >
> > > 1. Although pose-free GNVS methods can be applied to GNVS tasks with known GT poses, the current state-of-the-art methods, such as LEAP or PF-LRM, struggle to match the accuracy and efficiency of contemporaneous GNVS methods with known GT poses. These pose-free methods mainly compare themselves with known GT pose methods like pixelNeRF (CVPR 2021).
> > >
> > > 2. Pose-free GNVS methods are currently limited to simple object-level datasets (e.g., DTU or Omniobject3D). In contrast, our method not only conducts experiments at the complex scene level (RE10K) but also generalizes directly to object-level datasets (see Figure 1 and Table 1 in the rebuttal PDF).
> > >
> > > 3. The primary challenges of unposed GNVS methods and GNVS methods with known poses are different at this stage. Unposed GNVS methods need to address how to implicitly learn geometric relationships or recover camera poses without any known 3D priors. They often reduce task complexity through clever structured feature representations (e.g., Learned 3D Neural Volume in LEAP and Triplane in PF-LRM), but this comes at the cost of reduced model generalization. On the other hand, GNVS methods with known poses focus more on generalization ability and handling larger baselines. Our epipolar-free method is designed to enhance performance on larger baselines: eFreeSplat significantly outperforms SOTA GNVS methods in challenging areas such as regions with smaller viewpoint overlaps.

---

> > > > ### Comment · Reviewer_1Kz3 · 2024-08-12
> > > >
> > > > Thanks for the detailed explanation.
> > > >
> > > > I can understand the argument of the reviewer. However, I still cannot agree with the authors that categorizing the methods into epipolar-free and geometry-free is appropriate. In my perspective, epipolar-free method is a subset of geometry-free methods. Thus, I am not able to agree that splitting unposed methods like SRT to geometry-free means it is not epipolar-free. The authors replied "*'epipolar-free' means that epipolar line sampling is not required*", and SRT, GS-LRM, indeed don't use epipolar line sampling.
> > > >
> > > > At the same time, I agree that SRT is geometry-free, but I don't agree GS-LRM is geometry-free. Moreover, I agree that unposed GNVS and posed GNVS are two tasks. However, whether the method is epipolar-free is orthogonal to whether the method uses GT poses, and it's very clear that the authors claimed "we are the first epipolar-free GNVS work".
> > > >
> > > > I realized this discussion can be a never-ending battle. So I would switch the context to my other comments that are not addressed. Is there any reply to my comments regarding the novelty of this work? And do you have any comments on why the performance gain is quite similar for data with overlap of 0.5 and 0.7?

---

> ### Author Response · Authors · 2024-08-13
>
> Thank you for your thoughtful response. Please find below our detailed comments on the novelty of our work and an analysis of the performance gain.
>
> ## **Novelty of the Work:**
>
> * **Epipolar-Free Approach to Address Overlap Challenges.**
> We have identified that mainstream multi-view GNVS methods, which rely on epipolar geometry, may encounter significant challenges in scenarios with large viewpoint overlaps. To address this, we explored the role of cross-view pretraining models that provide 3D prior knowledge for the GNVS task, and we introduced a cross-view mutual perception mechanism to partially mitigate the limitations of current methods.
>
> * **Iterative Cross-view Gaussians Alignment Module.**
> Beyond cross-view mutual perceptive pretraining, we have specifically introduced the Iterative Cross-view Gaussians Alignment module. We observed that relying solely on the CroCo pretraining model is insufficient to effectively address challenges such as ambiguity in GNVS, primarily due to the absence of precise local feature matching. Unlike previous generalizable NVS methods utilizing Gaussian Splatting (e.g., MVSplat), our approach circumvents the need for time-consuming and less generalizable structured feature representations, bypasses complex depth sampling processes, and enables efficient per-pixel Gaussian attribute prediction. The ablation experiments presented in Table 3 and Figure 6 demonstrate the critical importance of the Iterative Cross-view Gaussians Alignment module.
>
> **Our approach not only effectively manages scenarios where epipolar geometry may fail but also eliminates the time-consuming process of sampling along epipolar lines. This makes our method particularly efficient and effective for GNVS tasks, especially in scenarios with extensive viewpoint overlaps.**
>
> ## **Performance Gain Analysis:**
>
> With regard to Table 2 of the manuscript, the three subsets with overlaps below 0.7, 0.6, and 0.5 each represent challenging experimental conditions where epipolar geometry becomes unreliable, and it would be inaccurate to assert that one is inherently more challenging than the others. **Moreover, the performance gain across these subsets does not follow a linear progression.** The reasons are as follows:
>
> * **Distribution Gap Between Subsets and Training Set.**
> These three low-overlap subsets differ from the training set and constitute a small fraction of the test set. The majority of training data involves input viewpoints that overlap well above 0.7. In the test scenes, the ratio of scenes with overlaps greater than 0.7 to those below 0.7, 0.6, and 0.5 is 45:8:5:1. Thus, from a dataset distribution perspective, these three subsets exhibit significant variability, and the overlap size does not directly correlate with difficulty. This distribution gap explains why the performance gain remains similar across these subsets, as the model’s performance is more heavily influenced by distributional differences than by the specific overlap percentage.
>
> * **Non-linear Error Growth with Decreasing Overlap.**
> In 3D computer vision, local matching errors can propagate and affect the global output, meaning that overall error does not necessarily increase linearly as overlap decreases. This is because incorrectly matched features might be propagated as outliers into subsequent modules (especially in modules with global awareness, such as transformers), thereby amplifying their impact on overall performance. Once overlap falls below a certain threshold, traditional methods are significantly compromised. For instance, if the feature point matching error rate is too high, SFM algorithms may fail. Consequently, scenes with overlaps below 0.7–0.5 all represent scenarios where epipolar geometry is unreliable, and there is no strictly linear relationship for such "unreliableness". Thus, the similarity in performance gains for overlaps of 0.5 and 0.7 reflects the model’s consistent capability in managing these challenging scenarios where epipolar constraints are inadequate.
>
> Finally, thank you again for your excellent suggestions, which have helped us to think more deeply about the essence of performance improvement in our method. We will incorporate this analysis of the performance gain into the revised manuscript.

---

> ### Comment · Reviewer_1Kz3 · 2024-08-14
>
> Thanks for the detailed and helpful reply!
>
> The motivation of using epipolar-free methods makes sense to me. While this field is super competitive as there are many recent works, releasing unnecessary inductive bias on geometric designs is promising to me, especially when having large data. The competition in this field makes me a bit tired, as I can feel most of the works are a kind of incremental. I appreciate the effort to further improving Dust3r training pipeline on the GNVS task using the proposed epipolar-free method and the iterative cross-view gaussian alignment method.
>
> Regarding the performance analysis, maybe the reason for my previous question is the data distribution (45:8.5:1 as mentioned). The data amount with small overlap ratio, e.g. 0.5, can be too less, so the numbers are also noisy. I sincerely suggest the authors to find some other data that are more balanced to perform the analysis, e.g. DTU or DL3DV, where you can control the overlap ratio as the two datasets provide dense views. I won't request this experiment as the discussion period is coming to the end soon.
>
> At the same time, we still want to let the authors know I expect the prior works can be correctly discussed, especially removing some controversial words like first "the first epipolar-free GNVS method". After the discussion, I don't want to reject this paper because of the overclaiming problem as I saw the effort behind this paper.
>
> Thus, I would like to raise the score to **a neutral borderline** (with a score of 4.5, due to the uncertainty of how the over-claiming problem will be resolved) and I don't against accepting this paper. As there is no neutral borderline for the reviewer, I will just kept the current rating now, but I will let AC know if I keep the a neutral borderline until the end of discussion.
>
> **At the same time, if you have better wording for resolving the controversial claims, please let me know before the discussion period ends. If the revised version is satisfying to me, I will raise the score to a borderline accept.**

---

> > ### Author Response · Authors · 2024-08-14
> >
> > Thank you for your support of our work and for providing valuable suggestions. We have carefully considered the issues you raised and plan to revise them as follows:
> >
> > ### **Revision of Controversial Claims**
> > Upon careful review, the original claims regarding the novelty and contribution of our method were as follows:
> > * Our eFreeSplat represents a new **paradigm** for generalizable novel view synthesis.
> > *	To our best knowledge, **the first multiview GNVS paradigm** that operates without relying on epipolar priors.
> > *	a **groundbreaking** generalizable 3D Gaussian Splatting model designed for novel view synthesis across scenes, **free from the constraints of epipolar priors**.
> > *	has inspired our development of a **novel GNVS paradigm** that circumvents the dependence on epipolar priors.
> >
> > After an in-depth discussion during the rebuttal, we recognize the validity of your suggestions, which significantly aid in more accurately describing our contribution. We will revise the content as follows:
> >
> > *	Our eFreeSplat represents an innovative approach for generalizable novel view synthesis. Different from the existing pure geometry-free methods, eFreeSplat focuses more on achieving epipolar-free feature matching and encoding by providing 3D priors through cross-view pretraining.
> > *	The approach with novel insights into GNVS that operates without relying on epipolar priors in the process of multi-view geometric perception.
> > *	A novel generalizable 3D Gaussian Splatting model tailored for novel view synthesis across new scenes, designed to function independently of epipolar constraints that might unrelieble when large viewpoint changes occur.
> > *	has inspired our development of a novel GNVS method that circumvents the dependence on epipolar priors through data-driven 3D priors.
> >
> >
> > ### **Revision of Discussion on Related Works**
> > After thorough review, we acknowledge that the original manuscript's description of closely related work included the following content but lacked discussion on methods like GS-LRM, which do not require epipolar line sampling:
> >
> > *	SRT[1] is a geometry-free, generalizable NVS method that boldly eschews any explicit geometric inductive biases. SRT encodes patches from all reference views using a Transformer encoder and decodes the RGB color for target rays through a Transformer decoder
> >
> > After in-depth discussion in the Rebuttal, we agree that your suggestion is reasonable, and we will provide a more detailed description of related work in our revised version:
> >
> > *	SRT[1] and GS-LRM[2] are epipolar-free GNVS methods that boldly eschew any explicit geometric inductive biases. SRT encodes patches from all reference views using a Transformer encoder and decodes the RGB color for target rays through a Transformer decoder. GS-LRM's network, composed of a large number of Transformer blocks, implicitly learns 3D representations. However, due to the lack of targeted scene encoding, these methods are either limited to specific datasets or suffer from unacceptable computational efficiency and carbon footprint.
> >
> > *	Some pose-free GNVS methods[1][3][4] are also epipolar-free. These methods, lacking known camera poses, find it challenging to perform epipolar line sampling. They often reduce task complexity through clever structured feature representations (e.g., Learned 3D Neural Volume in LEAP[3] and Triplane in PF-LRM[4]), but this reduction comes at the cost of decreased model generalization. Different from the above methods, our proposed eFreeSplat focuses on data-driven 3D priors and does not require any time-consuming and complex structured feature representations, such as cost volumes.
> >
> >
> > As for the experiments on a more overlap-balanced dataset, due to time constraints, as you acknowledged, we regret that it is challenging to prepare such balanced data and train our model in time to include the analysis within the eFreeSplat work. Nevertheless, we commit to establishing such an experimental scene in the forthcoming revised draft and to eliminating the influence of data distribution gaps on the experimental results, thereby providing a more thorough analysis of our method's advantages in scenarios with significant viewpoint changes.
> >
> > **We will do our utmost in the revised version to resolve the controversial claims and address all the issues mentioned above. Once again, thank you for your professional feedback and evaluation of our work.**
> >
> > [1] Sajjadi M S M, Meyer H, Pot E, et al. “Scene representation transformer: Geometry-free novel view synthesis through set-latent scene representations” CVPR 2022.
> >
> > [2] Zhang K, Bi S, Tan H, et al. “GS-LRM: Large Reconstruction Model for 3D Gaussian Splatting” ECCV 2024.
> >
> > [3] Jiang, Hanwen et al. “LEAP: Liberate Sparse-view 3D Modeling from Camera Poses.” ICLR 2024.
> >
> > [4] Wang, Peng et al. “PF-LRM: Pose-Free Large Reconstruction Model for Joint Pose and Shape Prediction.” ICLR 2024.

---

> > > ### Comment · Reviewer_1Kz3 · 2024-08-14
> > >
> > > Thanks for your effort during the rebuttal period. I will raise the score.

---

### Official Review · Reviewer_AnyG · 2024-07-12

**Soundness:** 3
**Presentation:** 3
**Contribution:** 3
**Rating:** 5
**Confidence:** 3

**Summary:**

This paper addresses the task of 2-view generalizable novel view synthesis. It introduces a cross-view completion model as prior assistance and incorporates cross-view Gaussian alignment after predicting Gaussian attributes to enhance cross-view consistency.

**Strengths:**

1. The introduction of pretraining the cross-view completion prior for assisting the generalizable novel view synthesis task is an interesting and novel approach.
2. Extensive experiments, including comparisons with baseline methods such as pixelSplat/MVSplat, demonstrate that the proposed approach can generate higher-quality novel view images, particularly in scenes with smaller overlaps (Tab. 2, Fig. 5).

**Weaknesses:**

The ablation analysis in Tab. 3 does not clearly identify which module contributes the most to the performance improvement.

**Questions:**

1. Table 3 seems to indicate that all three modules are important. In the author's opinion, which module has the greatest impact on the performance improvement?
2. Does the iterative nature of the Cross-view Gaussians Alignment module slow down the training speed? Is choosing to iterate twice a trade-off between effectiveness and efficiency?
3. Is Cross-view Gaussians Alignment required during inference?

**Limitations:**

The author provides a detailed discussion of limitations and social impacts in the paper.

---

> ### Author Rebuttal · Authors · 2024-08-07
>
> ### **Q1. The most influential module in Table 3**
> The results in Table 3 show that the absence of the epipolar-free cross-view mutual perception results in the most significant performance decline. This highlights the crucial role of the pretraining and network structure of Croco in providing 3D priors and cross-view features. Additionally, as illustrated in Fig. 6, the exclusion of this module leads to noticeable artifacts in both novel view images and depth maps, underscoring its importance. We will add this analysis to the ablation study  section in the paper.
>
> ### **Q2. Supplementary analysis of the Iterative Cross-view Gaussians Alignment module**
> While the iterative nature of the Cross-view Gaussians Alignment module does slightly reduce the rendering speed during training or inference, it substantially enhances the reconstruction metrics. The supplementary experiments with iterations set to 1-3, as presented in the table below, demonstrate that iterating twice strikes a balance between effectiveness and efficiency, providing significant improvements without excessively increasing computational demands. For details, please see **Table 2 and Figure 2 in the Rebuttal PDF**.
>
> ### **Q3. Whether Cross-view Gaussians Alignment required during inference**
> Yes, cross-view Gaussians Alignment is required and plays a crucial role during inference, and it does not require much time (less than 0.061s, as reported in Tab 1 of the manuscript). The scene representation is obtained from input perspective images through Epipolar-free Cross-view Mutual Perception and Cross-view Gaussians Alignment, which matches the scene representation trained by the single-scene optimized 3DGS method, allowing for direct inference of new viewpoints. As shown in Fig. 2 of the manuscript, the pipeline of eFreeSplat remains the same for both the training and testing sets: it first uses Croco-pretrained ViT to provide feature maps with 3D priors, then employs the Cross-view Gaussians Alignment module to acquire pixel-wise 3D Gaussian primitives for the 3D scene representation, and finally, the inference is performed using the original 3DGS tile-based rendering method.

---

> > ### Author Response · Authors · 2024-08-12
> >
> > Thanks so much again for the time and effort in our work. May I know if our rebuttal addresses the concerns? If there are further concerns or questions, we are more than happy to address them. Thanks again for taking the time to review our work and provide insightful comments.

---

### Official Review · Reviewer_MtQr · 2024-07-13

**Soundness:** 2
**Presentation:** 3
**Contribution:** 3
**Rating:** 4
**Confidence:** 3

**Summary:**

This paper introduces a robust pipeline for generalizable 3D Gaussian novel view synthesis, utilizing a cross-attention model trained on large-scale datasets. This approach enables the model to generalize effectively to new scenes without depending on epipolar geometry, which is commonly used in traditional methods. To address inconsistencies in depth scales, the pipeline features an iterative refinement process that adjusts the attributes of 3D Gaussians based on cross-view feature correspondences.

**Strengths:**

- The motivation is well-grounded. The paper identifies and tackles one of the core problems of the sparse 3D Novel view synthesis.

- The experiments show solid results on image pair settings, especially in sparse and non-overlapping scenarios. The evaluation convinces me of the usefulness of each component.

**Weaknesses:**

- Overall, the paper has good ideas, but the experiments are not enough to support their claim. The paper claimed to be a multiview generalizable novel view synthesis, but most of the experiments are done in an image pair setting, it should add more experiments in a sparse-view setting and compare with other sparse-view Novel view synthesis methods on datasets like mipnerf360.
- 59 "obtain the warped features for each view based on the predicted depths via U-Net" and formula (5) seems similar to the the coarse-to-fine cost volume formulation utilized in the MVSNet. Could you elaborate on how these two are different?

**Questions:**

- In formula (7) , the C is used without definition. I assume it is the dimensionality of the feature of 3D Gaussian primitive? It should be made clearer.
- Since you are using a Croco pre-trained model could you give some metrics on 512 x 512 resolution?
- Given the significant memory demands of the global alignment process in DUSt3R, could you detail the memory requirements for your method? It would be helpful to understand if similar memory constraints apply to your approach.

**Limitations:**

The author have made a commendable effort in acknowledging both the technical limitations and potential negative societal impacts of their method.

---

> ### Author Rebuttal · Authors · 2024-08-07
>
> ### **Weaknesses 1. Missing experiments in a sparse-view setting and comparison with other sparse-view methods**
> We thank you for identifying the novelty of our idea. Nevertheless, maybe we did not illustrate the difference between GNVS and sparse-view NVS clear enough and make you miss the focus of our experiments. Please allow us to reiterate the concept of the generalizable novel view synthesis (GNVS) task:
> The GNVS task aims to render new viewpoint images by leveraging the generalization ability of cross-scene 3D representations. When encountering new scenes not present in the training set, this process requires no additional single-scene optimization, achieving rendering through a single network feed-forward pass. Therefore, sparse-view novel view synthesis and GNVS are distinct tasks. The methods and datasets we compare in our experiments are commonly used settings in the GNVS task.
> Moreover, eFreeSplat is theoretically straightforward to extend to multi-view inputs. Our choice of datasets and dual-view input settings is solely for fairer comparisons with current GNVS methods. For instance, recent works such as pixelSplat and MVSplat also use dual-view input in their experimental settings, with their datasets being RE10K and ACID. Through fair experiments, we have validated our method's generalization capability and its advantages in reconstructing challenging regions.
>
> ### **Weaknesses 2. Differences between Eq. (5) and cost volume construction method in MVSNet**
> Here, we provide a detailed explanation of the differences between Equation (5) and the coarse-to-fine cost volume construction method in MVSNet.
>
> Equation (5) represents a commonly used warped transformation formula in multi-view geometry. Unlike MVSNet, which requires sampling along the canonical view ray N times to obtain N depth values $\[d_{i}\]_{i=1}^{N}$  when constructing the cost volume, our method does not require multiple depth direction samples. Specifically, in each iteration, our method directly predicts the coarse depth value $d$ using a U-Net network (as detailed in Equation (4)), rather than sampling each depth plane to calculate the cost volume as MVSNet does.
>
> The intuition behind this approach stems from the 3D prior knowledge provided by the CroCo pretrained model, which replaces the multiple depth sampling process required in the MVSNet cost volume. By leveraging CroCo's 3D priors, we can effectively obtain multi-view feature point matching relationships in each iteration, simplifying computation and improving efficiency.
>
> ### **Q1. Missing definition**
> Thank you very much for your thorough review and valuable suggestions. We sincerely apologize for the omission of the definition of $C$ in Equation (7). Your assumption is indeed correct: $C$ represents the feature dimension of the 3D Gaussian primitives. We will include a clear definition of $C$ in the revised manuscript.
>
> ### **Q2. More experiments on 512 x 512 resolution**
> Thank you for your valuable feedback. The baseline methods use the RE10K and ACID datasets, which have a resolution of 360 x 640, lower than 512 x 512. For a fair comparison, we did not conduct experiments at other resolutions that were different from the baselines and other popular methods.
>
> According to our research, current 3DGS-based GNVS methods all use fixed, lower-resolution datasets for testing. Although high resolution and variable resolution are not the focus of this study, they could indeed make models more applicable in real-world scenarios, so we have included them in our future work plans. Thank you again for your suggestion.
>
> ### **Q3. The memory requirements**
> Thank you for your valuable suggestion. We have added ablation experiment results for iteration counts of 1-3, including comparisons of memory usage, rendering speed, and reconstruction metrics. For details, please see **Table 2 and Figure 2 in the Rebuttal PDF**.
>
> The experiments show that while our method's memory usage varies with different iteration counts, the overall memory demand remains relatively low. Compared to DUSt3R, our method requires less memory and achieves better rendering efficiency and reconstruction accuracy with two iterations.

---

> > ### Author Response · Authors · 2024-08-12
> >
> > Thanks so much again for the time and effort in our work. May I know if our rebuttal addresses the concerns? If there are further concerns or questions, we are more than happy to address them. Thanks again for taking the time to review our work and provide insightful comments.

---

### Official Review · Reviewer_w4rj · 2024-07-13

**Soundness:** 3
**Presentation:** 3
**Contribution:** 3
**Rating:** 7
**Confidence:** 5

**Summary:**

This work presents eFreeSplat, a model to address generalizable novel view synthesis without relying on epipolar prior. Specifically, it extracts the cross-view mutual perception by leveraging a pre-trained CroCo model. It then improves the alignment of multi-view Gaussians by using an iterative updating strategy. Finally, novel views are rendered by an off-the-shelf 3DGS renderer. Experiments on two benchmarks, RE10K and ACID, demonstrate the effectiveness of the introduced eFreeSplat.

**Strengths:**

* The motivation of using the pre-trained model to address the non-overlapping and occluded regions is interesting.

* Extensive experiments showcase that eFreeSplat achieves significantly better results on scenes where input views contain less overlap, which is well-aligned with the motivation.

* The paper is well-written and easy to follow.

**Weaknesses:**

* Performances regarding cross-dataset generation. Similar to the cross-dataset experiments shown in MVSplat, it would be interesting to see how eFreeSplat performs when trained on RE10K but tested on ACID and DTU. This experiment will help us better understand how such a data-driven approach generalizes across different data distributions.

* It would be better to provide a more detailed analysis of the Iterative Cross-view Gaussians Alignment module. For example, showcase the depth maps of both input views when setting the iteration to 1, 2, and 3. This would be helpful for verifying whether the iterative solution can really help align the depth scale across multiple views.

* Will it be beneficial to perform a fine-tuning of the CroCo model? As mentioned in the failure case, eFreeSplat performs worse on cases with significant overlaps, potentially because CroCo is trained on image pairs with slight overlaps. Can these failure cases be addressed by fine-tuning CroCo on the RE10K training set? Similarly, as reported in Tab. 3, the performance of eFreeSplat w/o pre-training weights is much worse compared to other state-of-the-art models, while MVSplat achieves reasonably good results even when training from scratch. Can this gap be reduced by pre-training the backbone model using the RE10K training set?

**Questions:**

Kindly refer to the [Weaknesses]

**Limitations:**

Kindly refer to the [Weaknesses]

---

> ### Author Rebuttal · Authors · 2024-08-07
>
> ### **Weaknesses 1. Performances regarding cross-dataset generation**
> Thank you for your valuable suggestions on our work. We fully agree with your view on the importance of cross-dataset testing and, following your advice, have added cross-dataset test results on the DTU dataset. For specific experimental results, please see **Table 1 and Figure 1 in the Rebuttal PDF**.
>
> These tests further validate eFreeSplat's generalization ability across different data distributions. The results demonstrate that even though training was conducted on the RE10K dataset, eFreeSplat performs excellently on the DTU dataset, which further confirms the robustness and generalization performance of our method.
> ### **Weaknesses 2. Supplementary analysis of the Iterative Cross-view Gaussians Alignment module**
> Thank you for your valuable suggestions. We have added ablation experiments for iteration counts of 1-3 and presented the corresponding depth maps based on your feedback. For specific experimental results, please see **the Table 2 and Figure 2 in the Rebuttal PDF**.
>
> The results indicate that setting the iteration count to 2 achieves the best balance between reconstruction accuracy and rendering efficiency. When the iteration count is set to 3, there is no significant improvement in image reconstruction metrics, which may be due to training overfitting and the subsequent iterations losing the 3D perception provided by the original CroCo features.
>
> ### **Weaknesses 3. Fine-tuning of the CroCo model**
> Thank you for your valuable suggestions. We conducted relevant Experiments A and B regarding fine-tuning the CroCo model using the RE10K dataset. Experiment A involved fine-tuning the CroCo pretrained weights with the RE10K training set, while Experiment B involved training CroCo directly with the RE10K training set without loading the pretrained weights. Finally, we retrained eFreeSplat using the new pretrained weights. The CroCo pretraining for Experiments A and B was performed on 2 RTX 4090 GPUs, with total iterations of 4000 and 6000, respectively (due to time constraints), learning rates of 2e-5 and 2e-4, and a batch size of 12. The viewpoint overlap was set the same as in the main model training.
> For the quantitative and qualitative results of Experiments A and B, please see **Table 3 and Figure 3 in Rebuttal PDF**.
>
> The results indicate that pretraining the backbone model on the RE10K training set effectively addresses the model's poor performance in low-overlap scenarios. However, in the RE10K test set, Experiment A's reconstruction metrics were slightly lower than those of the original model, which may be due to insufficient training iterations. We will further investigate the positive impact of fine-tuning the CroCo pretrained model on novel view synthesis and 3D reconstruction in future work.

---

> ### Author Response · Authors · 2024-08-12
>
> Thanks so much again for the time and effort in our work. May I know if our rebuttal addresses the concerns? If there are further concerns or questions, we are more than happy to address them. Thanks again for taking the time to review our work and provide insightful comments.

---

### Author Rebuttal · Authors · 2024-08-07

We thank the reviewers for their thoughtful feedback! We are encouraged that they found our motivation for using a pre-trained model to address non-overlapping and occluded regions interesting and well-grounded (w4rj, MtQr, 1Kz3). They also praised our paper's well-written presentation (w4rj) and the novelty of introducing pretraining the cross-view completion prior for assisting in the generalizable novel view synthesis task (AnyG). Additionally, they appreciated the beneficial improvements demonstrated through extensive experiments, particularly in scenarios with sparse and non-overlapping input views (w4rj, MtQr, AnyG). Here, we address the concerned questions below and will incorporate all responses in the revision.

---

### Comment · Area_Chair_sTgg · 2024-08-10

Hi reviewers,

Thank you for your hard work in reviewing the paper!
Please check out the authors' responses and ask any questions you have to help clarify things by Aug 13.

--AC

---

### Decision · Program_Chairs · 2024-09-25

**Decision:**

Accept (poster)

**Comment:**

This paper receives borderline reviews. It gets 1x accept, 2x borderline accepts and 1x borderline reject. The reviewers agree that the motivation of using the pre-trained model to address the non-overlapping and occluded regions, and the idea of not using epipolar geometry priors are interesting and well-grounded. Extensive experiments, including comparisons with baseline methods such as pixelSplat/MVSplat, demonstrate that the proposed approach can generate higher-quality novel view images, particularly in scenes with smaller overlaps. The major concern of Reviewer MtQr who gave a borderline reject is that the experiments do not seem enough to support the claims of the paper. This concern seems to be addressed by the authors during the rebuttal. It was pointed out in the rebuttal that "maybe we did not illustrate the difference between GNVS and sparse-view NVS clear enough and make you miss the focus of our experiments". The weaknesses mentioned by other reviewers are mostly clarifications on the experiments, which seem to have been well-addressed during the rebuttal and discussion phase. On the consideration that the strengths outweigh the weaknesses of the paper, the meta-reviewers decide to accept the paper.